# Systematic Review on CAR-T Cell Clinical Trials Up to 2022: Academic Center Input

**DOI:** 10.3390/cancers15041003

**Published:** 2023-02-04

**Authors:** Valentine Wang, Mélanie Gauthier, Véronique Decot, Loïc Reppel, Danièle Bensoussan

**Affiliations:** 1CNRS, Lorraine University, IMoPA, F-54000, Nancy, France; 2Cell Therapy and Tissue Bank Unit, CHRU Nancy, F-54000, Nancy, France

**Keywords:** clinical trial, Chimeric Antigen Receptor T cell, academic center, regulation

## Abstract

**Simple Summary:**

The development of CAR-T cell clinical trials has accelerated over the last two decades. These trials, collected on Clinicaltrial.gov until 2022, come mainly from the United States (*n* = 377) and China (*n* = 636), while Europe (*n* = 58) remains far behind these two leading countries. The aim of our analysis of clinical trials was to provide an overview of the characteristics of these trials, such as conditions, targets, phase status, and especially, pointing out the nature of the academic or industrial investigator. We have highlighted that poorly developed academic and industrial collaborations in Europe could be one explanation of the delay of Europe to bring a CAR-T cell product to market, compared to the leading countries. Moreover, regulatory and financial differences also come into play. A valorization of the development of these advanced therapy medicinal products as well as the provision of financial support would accelerate the process in Europe.

**Abstract:**

The development of Chimeric Antigen Receptor T cells therapy initiated by the United States and China is still currently led by these two countries with a high number of clinical trials, with Europe lagging in launching its first trials. In this systematic review, we wanted to establish an overview of the production of CAR-T cells in clinical trials around the world, and to understand the causes of this delay in Europe. We particularly focused on the academic centers that are at the heart of research and development of this therapy. We counted 1087 CAR-T cells clinical trials on ClinicalTrials.gov (Research registry ID: reviewregistry1542) on the date of 25 January 2023. We performed a global analysis, before analyzing the 58 European trials, 34 of which sponsored by academic centers. Collaboration between an academic and an industrial player seems to be necessary for the successful development and application for marketing authorization of a CAR-T cell, and this collaboration is still cruelly lacking in European trials, unlike in the leading countries. Europe, still far behind the two leading countries, is trying to establish measures to lighten the regulations surrounding ATMPs and to encourage, through the addition of fundings, clinical trials involving these treatments.

## 1. Introduction

Chimeric antigen receptor T cells, known as CAR-T cells, have generated extraordinary results in phase I/II clinical trials in the treatment of CD19+ B-cell hematological malignancies. The principle is based on the genetic modification of the patient’s immune T cells by transferring a transgene coding for a chimeric receptor. This receptor recognizes antigens present on the surface of targeted tumor cells, regardless of any major histocompatibility complex (MHC) restriction, leading to their destruction. CARs are usually composed of an extracellular domain—a single-chain variable fragment (scFv) of a monoclonal antibody implicated in the recognition of the target cell antigen—and an intracellular domain responsible for the activation and the function of T cells [1]. Different generations of CAR-T cells have been raised, depending on the composition of the intra-cellular domain: (i) first generation, CD3z chain [2]; (ii) second generation, CD3z chain and a co-stimulation domain such as CD28, 4-1BB or OX40 [3,4]; and (iii) third generation, CD3z chain and two co-stimulation domains. A fourth generation called TRUCKS (T cells Redirected for antigen-Unrestricted Cytokine-initiated Killing) has been recently developed, where the transgene coding for a second-generation CAR-T is completed with a gene coding for a cytokine such as IL-12 or IL-15, for example [5].

Clinical trials for CD19 CAR-T cells have shown a high level of complete or partial remissions in patients with poor prognoses. The results from the ELIANA and ENSIGN studies [6] showed a complete remission of 67% at 3 months in patients with acute lymphoblastic leukemia (ALL), which was maintained in almost 40% of patients after a median follow-up of 9 months. For non-Hodgkin lymphoma (NHL) patients, with a minimum of 6 months of follow-up, the objective response rate was 82%, with a 54% complete remission rate in 101 patients (ZUMA-1 study) [7]. The median overall survival rates were 78% at 6 months and 52% at 18 months. These significant results concern patients who had already received several lines of treatment; however, they were associated with severe adverse effects, especially cytokine release syndrome (CRS), an over-activation of the immune system, and neurotoxicity. Some of these effects are responsible for significant morbidity and mortality. The ELIANA and ENSIGN studies reported a major CRS and severe neurological toxicity in 77.2% of patients (ELIANA). The ZUMA-1 study observed a grade >3 CRS in 13% of patients, responsible for the death of 2 patients, and neurotoxicity in 28% of patients. Adverse effects have sometimes led to the discontinuation of a clinical trial. For example, Juno Therapeutics stopped its phase II ROCKET trial in ALL B in 2017 following five deaths caused by cerebral oedema. The experience gained from the clinical data has led to recommendations for the management of these events (tocilizumab availability and intensive care unit), to reduce their severity and increase patient survival. The use of a suicide gene, currently being tested in clinical trials (ClinicalTrials.gov Identifier: NCT02761915, NCT03373071 and NCT03373097), allows control of the CRS in cases of excessive over-activation. Nevertheless, despite these results and an effective management of adverse effects, more than half of patients will experience a relapse. In fact, according to data from follow-up studies, between 30-50% of patients who have been in remission are found to relapse within one year of the infusion of anti-CD19 CAR-T cells [8]. Cases of tumor escape have also been observed due to the absence or loss of CD19 expression by the tumor cells. To overcome this escape, other models with multispecific CAR-T cell are studied to decrease the risk of relapse most commonly by targeting CD19 and/or CD20 and/or CD22 in B cell malignancies. Moreover, it is still difficult to transpose this therapy to solid tumors. These involve numerous barriers of physical (access to the tumor), immunological (immunosuppressive environment induced by the tumor) and tumor targeting specificities, with an “off target” effect observed on healthy tissues. The development of this therapy in solid cancers is a major challenge for academic research teams and the pharmaceutical industry.

The first CAR-T cells were commercialized in 2017 as Yescarta^®^ (axicabtagene ciloleucel) from Kite/Gilead (Foster City, California) [7,9] and Kymriah^®^ (tisagenlecleucel) from Novartis (Basel, Switzerland) [6] following U.S. Food and Drug Administration (FDA) authorization under the Advanced Therapy Medicinal Product (ATMP) status (Figure 1A). Marketing authorizations were then obtained for (i) the anti-CD19 Tecartus^®^ (brexucabtagene autoleucel) from Kite/Gilead (Foster City, California) in 2020 for relapsed or refractory (r/r) mantle cell lymphoma and for adult patients with r/r B-cell precursor ALL, and (ii) for the anti-B-cell maturation antigen (BCMA) Abecma^®^ (idecabtagene vicleucel) from Celgene (Summit, New Jersey) in 2021 for the treatment of adult patients with r/r multiple myeloma after four or more prior lines of therapy, including an immunomodulatory agent, a proteasome inhibitor and an anti-CD38 monoclonal antibody. At the end of 2021, another anti-CD19 CAR-T cell, Breyanzi^®^ (lisocabtagene maraleucel) from Juno Therapeutics, Inc., a Bristol-Myers Squibb Company (Seattle, Washington), was granted authorization for the treatment of adult patients with r/r large B-cell lymphoma after two or more lines of systemic therapy, including diffuse large B-cell lymphoma (DLBCL), high-grade B-cell lymphoma, primary mediastinal large B-cell lymphoma, and follicular lymphoma grade 3B. At the beginning of 2022, Breyanzi^®^ was under review in Europe, Switzerland and Canada. Soon after, an anti-CD38 CAR-T cell, Carvykti™ (ciltacabtagene autoleucel) was authorized by the FDA in r/r multiple myeloma. As can be seen in Figure 1A, the development of CAR-T cells first began in academic centers before licensing to a pharmaceutical company, either directly (Novartis for Kymriah^®^) or through the creation of a start-up (Kite Pharma/Gilead for Yescarta^®^) (Foster City, California).

The production of CAR-T cells in an autologous context was not easy to implement for pharmaceutical companies used to producing batches of medicinal products from one batch of raw material. For autologous CAR-T cell generation, each raw material is for one batch of ATMP dedicated to the same patient. This led to a very intricate organization, requiring the participation of the academic leukapheresis center, the academic cell therapy unit, transport companies, and the academic hospital pharmacy (Figure 1B) [11,12]. As these key steps, at least, are not under industrial control, pharmaceutical companies insisted that each academic center observed specific rules and procedures to validate the opening of the center [13]. Such an intricate organization presents different drawbacks, such as (i) the complete production time (between 17 to 22 days according to the manufacturers), requiring production platforms as close as possible to the collection centers, to reduce this delay; (ii) the high production capacity of the pharmaceutical company platforms, to be able to answer the increasing medical need; (iii) an impact on the final cost of the CAR-T cells, which is very high, limiting the dissemination of this treatment all over the world. This final cost is partly related to this organization, for a pharmaceutical company that needs to ensure complete control of the ATMP production process. On the other side, academic centers are used to producing cells in an autologous context with a dedicated organization for each patient, as this is already performed around the world for hematopoietic stem cell transplantation (HSCT).

In this systematic review, we present the state-of-the-art clinical trials involving CAR-T cells in the world in 2022, before performing a comparative analysis between European trials and the world leaders (China and USA). Finally, we have attempted to analyze the modes of CAR-T manufacturing carried out in European trials led exclusively by academic centers. The results allow us to discuss the reasons for Europe’s delay and the role of academic centers in the production of CAR-T cells, particularly autologous CAR-T cells.

## 2. Materials and Methods

This systematic review follows PRISMA guidelines 2020 and is registered on Research Registry as “reviewregistry1542”. The clinical trials were collected from the public Clinicaltrials.gov database. The following search was performed: “chimeric antigen receptor OR CAR-T cell OR CAR-T cells OR CAR-T cell” [14]. Only “interventional studies” were selected, regardless of the clinical indication. The following information was collected for each study: investigators, sponsors and collaborators; country of origin of the investigator; status; phase; indication; biological target; CAR sequence generation and design; cell source; and steps of production. The data collected on Clinicaltrials.gov were supplemented by published articles and journals that have studied the design of CAR-T cells produced in clinical trials, as well as information made public by pharmaceutical companies (pharmaceutical industry pipelines). Because this analysis was based on the baseline characteristics of the clinical trials, and not on the clinical results, biases may be related to the method of selection of the trials and the information published in the database. Missing data from the database were mentioned as “Not specified”. The software Microsoft excel was used to create figures. Clinical trial selection and data collection process were assessed by the first author and analysis were assessed by all authors. The data on the international trials were studied in a global way in the first step; then, in the second step, the data of the European trials were analyzed more precisely, in comparison with those of the international trials. All trials were checked one by one, no automation tools were used.

## 3. Results

### 3.1. Study Selection

At the date of the 25 January 2023, 1205 results were generated. Studies were excluded when they started on or after 1 January 2023 (*n* = 31) and when CAR-T cells were not involved (*n* = 85). After the first selection, 1087 interventional clinical trials were reviewed in our work for a global analysis. The second selection on European trials (*n* = 58) was done for a comparative analysis with non-European clinical trials (*n* = 1029).

### 3.2. Geographic Situation, Clinical Trial Status, and Phases of the Clinical Trials

The geographical distribution of the trials was based on the country of origin of the investigator. As shown in Figure 2A, most of the trials are in Asia (645 studies, 59.3% of the trials) and North America (380; 35%), far ahead of the other continents. Asia is dominated by China (645; 58.5%), while America is dominated by the United States (USA, 377; 34.7%). These two countries are thus the main competitors in the development of CAR-T cells. As shown in Figure 2B, among the 1087 trials, 117 have not started recruitment (10.8%), 583 are in the process of recruitment (53.6%), 110 are currently active (10.1%), 51 have been completed (4.7%), 56 have been discontinued (either suspended, terminated or withdrawn; 5.2%) and 170 trials are classified as “unknown status” (15.6%) due to lack of updates on the site from the authors or because the theoretical end of study deadline has passed.

As expected, most of the studies are early-phase-development clinical trials (Figure 2C), with only 73 in phase II (6.7%), 6 in phase II/III (0.6%), and 7 in phase III (0.6%). Fifty-seven clinical trials did not mention which phase they were related to (5.2%). We counted many more clinical trials than the study by Hartmann [15]: 1087 versus 188 in 2016. While the USA were the world leader in 2016, Asia has now moved far ahead into first place. As in 2016, most of the trials are currently in early-phase clinical development (I and I/II).

### 3.3. Analysis of the Clinical Trials According to Their Academic or Pharmaceutical Company Sponsorship

The sponsors were collected for each trial according to whether they were academic or industrial (Figure 2D). On the Clinicaltrials.gov site, we observed that pharmaceutical or biotechnology companies were the sponsors of 258 studies (23.7%), a collaboration of academic centers and industrial partners were the sponsors of 369 studies (33.9%), and exclusive academic sponsorship of CAR-T cell clinical trials was observed for 385 studies (35.4%).

We noticed that there were nearly as many trials sponsored by an academic center alone as there were by a collaboration between an academic center and an industrial group. Moreover, 62 trials (5.7%) were sponsored by an academic center supported by a government institution (mostly represented by the National Institute of Health (NIH) in the USA, or by ministerial departments in China) and 13 studies (1.2%) by the collaboration of an academic center, a pharmaceutical company and a government institution. Altogether, academic center sponsorship is implicated in a large majority of the clinical trials on CAR-T cells, testifying to the strong contribution of academic centers to the innovation in this field. As expected, those trials implicating academic centers alone or in association are mainly early-phase clinical trials (Phase I and Phase I/II for 86.8%, Figure 2E). On the other hand, industrial groups alone sponsored 23.7% of the trials. The pharmaceutical groups included in at least 10 clinical studies of CAR-T cell therapy are: PersonGen (Suzhou, China), BioTherapeutics (Suzhou, China), Hebei Senlang Biotechnology(Shijiazhuang, China), Yake Biotechnology (Shanghai, China), Novartis (Basel, Switzerland), Kite/Gilead (Foster City, California), Chongqing Precision Biotech (Chongqing, China), CARsgen Therapeutics (Shanghai, China), Beijing Immunochina Medical Science & Technology (Beijing, China), Nanjing Legend Biotech (Nanjing, China), Celgene (Summit, New Jersey), Hrain Biotechnology (Shanghai, China), Shanghai Unicar-Therapy Bio-medicine Technology (Shanghai, China), and iCell Gene Therapeutics (New York, NY, USA).

In summary, we observed that USA and China have remained the leaders in CAR-T cell therapy for the two last decades in terms of the number of clinical trials. More than 10% of the studies are not yet active (awaiting for recruitment) illustrating the number of studies that will begin in the near future. The majority are in recruitment or active (63.7%) with data expected in the next few years. We also observed that most of trials with unknown status are Chinese. Regarding the nature of the investigator, the distribution is homogeneous between the choice of a sponsor from an academic center, from industry or from a collaboration between the two. In the USA, the NIH is also involved in the implementation of these trials, mainly by supporting an academic center. Finally, the majority of trials involves a pharmaceutical company (as a unique sponsor or in collaboration). There were three phase-II/III trials and seven phase-III trials. Although one phase IV academic study was identified, it had been suspended. After this general review of the 1087 international clinical trials, we will perform a comparative analysis between these trials and the European trials.

### 3.4. CAR-T Cell Clinical Trials in Europe Up to 2022

We will now analyze the 58 clinical trials investigated by Europe in 2022 (Figure 3A). These trials have mainly been carried by the United Kingdom (17 studies) and Germany (9), followed by Italy (6), Belgium and Spain (4 each), and Sweden and Israel (3 each). The Czech Republic, Switzerland, Russia, and France have each investigated two studies. Finally, Lithuania, Austria, Finland, and Netherlands have each investigated one study. We have analyzed the European trials in the same way as the international trials and compared them with the data analysis of the international non-European trials.

The status of European trials until the end of 2022 included six studies awaiting enrolment, thirty-four studies that were enrolling, nine active studies, seven completed studies, and four discontinued studies (Figure 3A). We observed that European trials are following the same trend as international trials, with around 10% of trials awaiting for recruitment and almost 74% of trials that are recruiting and active. All European trials have an updated status, which is not the case for 170 international trials, the vast majority of which originate in China. An analysis of the phases of the European trials shows that they mostly concern phase I and II trials, just as with the international trials (Figure 3B). Indeed, 24 studies concern phase I trials and 23 studies are phase I/II trials, constituting 81% of European trials. Europe then has nine phase II studies, one phase II/III study, and one phase III study. Thus, early phase trials remain in the majority, both internationally and in Europe. The distribution of trials according to the nature of the sponsor is shown in Figure 3C, where 34 trials are sponsored by an academic center, 21 by a pharmaceutical company, and only 3 studies by an academic-industry collaboration; meanwhile, for international trials, academic-industrial collaborations have as many studies as academic centers (366 and 351 trials, respectively) and more than industry (237 studies). We can see that academic and industry collaboration is not yet widespread in Europe, unlike international trials, and especially those coming from the USA and China. Both of these countries also benefit from government-wide collaboration. The NIH is mostly involved in granting funds for the American trials. Meanwhile, European clinical trials are mostly sponsored by an academic center alone or by an industrial company. The European companies involved are: Miltenyi Biotec (Bergisch Gladbach, Germany), Autolus Limited (London, United Kingdom), Cellectis SA (Paris, France), Celyad Oncology SA (Mont-Saint-Guibert, Belgique), BioNTech Cell & Gene Therapies GmbH (Mainz, Germany), Cellex Patient Treatment, Pharmalog (Dresde, Germany), Novartis Pharmaceuticals (Basel, Switzerland) and AGC Biologics (Søborg, Danemark). Moreover, AFA Insurance (Sydney, Australia) is involved for Swedish clinical trials. Regarding the implementation of academic trials, some centers, which are mostly university hospitals, benefit from the support of research institutions and charitable foundations but also from the European Union. Figure 3D shows that 45 European clinical trials studied hematological malignancies and 12 trials studied solid tumors. We can see that European trials are mostly focused on hematological malignancies, while international non-European trials tend to study solid tumors to a greater extent. Some trials include both types of cancers.

Figure 4A shows the conditions treated among the hematological malignancies. We can see that the majority of both international and European trials are shared between all lymphomas (278 and 21 studies, respectively), acute or chronic LL (207 and 15 studies, respectively), and multiple myeloma (139 and 8 studies, respectively). With regard to the solid tumors in Figure 4B, the vast majority of international trials focus on liver cancer (51), glioblastoma (48), pancreatic cancer (42), and lung cancer (33). Only glioblastoma is studied (two) among these three indications, additionally with colorectal cancer (three), neuroblastoma (two), and then prostate cancer, malignant pleural mesothelium, and head and neck cancer (one study each). The main targeted antigens in hematological malignancies in most of the trials, both for international and European studies, were CD19 (354 and 28, respectively) and BCMA (101 and 4, respectively), which is the targeted antigen in the new CAR-T cell therapy that have recently received marketing authorization (Abecma^®^ from Celgene (Summit, New Jersey)). It can be noted that the European trials for hematological malignancies are studying new antigenic targets; for example, SLAMF7 (multiple myeloma), CD44v6 (acute myeloid leukemia and multiple myeloma), or TRBC1 (large B-cell lymphoma). Regarding the targets chosen to treat solid tumors (Figure 4D), the distribution for international trials is between mesothelin (34), MUC1 (27), and GPC3 (27), whereas European trials have chosen other targets, such as NKG2D (4 studies in colorectal cancer and liver cancer), GD2 (2 studies in neuroblastoma), FAP (one study in malignant pleural mesothelioma) and T4 (one study in thyroid tumor) as innovative targets.

### 3.5. Academic Production of CAR-T Cells in European Clinical Trials

Among the 34 academic clinical trials reported in Europe in this retrospective analysis, we report a diversity within the manufacturing protocols of the CAR-T cells. The heterogeneity of manufacturing protocols includes variability in the source of the starting material, the cell activation mode, the genetic modification strategies, the expansion media and the culture vessels.

To date, T cells collected by apheresis remain the most common starting material for CAR-T cell products (ClinicalTrials.gov Identifier: NCT02893189). Another strategy aims to use other types of cells, such as virus-specific cytotoxic T cells (ClinicalTrials.gov Identifier: NCT01195480), invariant Natural Killer T cells from Baylor College Medicine (ClinicalTrials.gov Identifier: NCT03294954), or Natural Killer cells from MD Anderson Cancer Center (ClinicalTrials.gov Identifier: NCT03056339), as a raw material to generate CAR-T cells. Autologous T cells collected by apheresis are currently the most widely used method in CAR-T cell production protocols. Whether an enrichment process of T cells is performed or not, a T cell activation step is essential for adequate transduction and expansion of CAR-T cells. To reach this goal, T cells are activated via polyclonal stimulation using anti-CD3 and anti-CD28 antibodies. Most of the manufacturing protocols use paramagnetic beads coated with these antibodies, such as Dynabeads, providing an appropriate activation of T cells. Recently, stimulation reagents such as Transact (Miltenyi Biotec, Bergisch Gladbach, Germany) have been employed, which uses humanized anti-CD3 and anti-CD28 antibodies conjugated to a colloidal polymeric nanomatrix for reliable activation of human T cells, whilst preserving high T cell viability and optimal immune function. After activation, T cells are genetically engineered to express CAR molecules. In our study, we noticed that CAR-T cells are generated by viral transduction (gamma-retrovirus or lentivirus) in most clinical trials, leading to a permanent CAR expression. A few are produced using mRNA or transposon technology, inducing transient CAR expression. Following transduction, the CAR-T cells undergo an expansion phase to reach the necessary clinical dose. This expansion step requires an appropriate expansion media to generate functional and non-exhausted T cells. Even though a significant number of clinical trials still use the usual IL-2 cytokine for T cell expansion, more and more studies have turned to a IL-7/IL-15 cytokine cocktail to promote more immature and less exhausted CAR-T cells [16,17]. The first manufacturing processes using gas-permeable culture bags, flasks, or bioreactors (G-Rex) relied on open, manual processing steps, which are extremely time-consuming, susceptible to operator-introduced errors and contamination, and not easily amenable to scale out [18]. Closed and semi-automated or automated systems, such as the CliniMACS Prodigy^®^ (Miltenyi Biotec, Bergisch Gladbach, Germany) [19] and the Lonza Cocoon^®^(Basel, Switzerland), have been developed to overcome these limitations. One major advantage of these systems is the ability to use them in less stringent cleanroom classifications. To date, CliniMACS Prodigy^®^ (Miltenyi Biotec, Bergisch Gladbach, Germany) is the most used device in several ongoing European clinical trials for the production of CAR-T cells. This automatic system is able to map all process steps, from cell preparation to harvest [20,21].

### 3.6. Two Different Examples of Academic Production in Europe

The academic production of CAR-T cells can be illustrated through the presentation of two academic clinical trials in the treatment of B-cell hematological malignancies (anti-CD19 CAR).

The first example is a Phase I pilot study in which the team of Manel Juan (Clinical Immunology Unit at SJD Barcelona Children’s Hospital, 08036, Barcelona, Spain) [22] is using the Miltenyi Prodigy^®^ (Bergisch Gladbach, Germany) to produce anti-CD19 ARI-0001 cells. This study confirms the feasibility of autologous CAR-T cells manufacturing according to GMP for the first time, using this device. Briefly, a selection of CD4/CD8 T cells is performed, and one fraction is activated by anti-CD3/CD28 antibodies coupled to magnetic beads and in the presence of IL-15 and IL-7 cytokines, while another is frozen for later use (quality control and in case of manufacturing failure). Transduction is performed with a lentiviral vector including sequences for the anti-CD19 scFV fragment, CD3z, and 4-1BB co-stimulatory domain. Cells are expanded up to the required doses: 1 × 10^6^ ARI-0001 cells/kg for lymphoblastic leukemia patients and 5×10^6^ ARI-0001 cells/kg for lymphoma patients. Cells are then cryopreserved. Overall, 28 products were generated using cells from 27 patients; one out-of-specification production was repeated. The 27 products met the specifications, before release for infusion into the patients. In terms of the total number of cells, the team was able to generate two doses of ARI-0001 per patient. The dose required in lymphoma was higher, with slower autologous T cell proliferation observed in these patients. A total of 9 doses/patient was obtained for each adult leukaemia patient and 25.4 doses/patient for the paediatric population; 2.5 doses/patient were obtained for the lymphoma patients. The average transduction of the 27 products was 30.6 ± 13.44% of cells transduced. Quality controls including sterility, cell viability, CD3+ cell count, transduction efficiency, potency assays, and presence of replicant competent virus copies were performed. In addition, the team performed a phenotypic analysis of the memory T-cell subsets. The authors observed that the expansion of cells appeared to be better with allogeneic cells from healthy donors, compared to those from diseased patients, which illustrates the issue of the quality of the raw material, especially if it is autologous, since patients who have already had several lines of chemotherapy may also have a reduced immune capacity.

The second example is a phase Ib/II study led by Sheba Medical Centre (Ramat Gan, Israel) with non-automated production of autologous CAR-T cells in a “small-scale” design [23,24]. Manufacturing of the 91 autologous CAR-T cell products was performed concomitantly for patients with ALL and NHL in the following steps: leukapheresis for patients who were not treated with fludarabine, and frozen for patients who had previously received it. Activation by the anti-CD3 monoclonal antibody OKT3 was performed, before retroviral transduction containing the scFV sequence CD19, CD3z and CD28 as a co-stimulatory molecule. The transduction efficiency was 73.5 ± 16.3% in ALL patients and 62.6 ± 19.6% in NHL patients at day 10. Expansion was performed in T175 flasks or GRex100 with IL-2 for about 10 days. After expansion, fresh non-cryopreserved CAR-T cells were infused into patients after a processing time of only 9 to 10 days. Quality controls including the cell count, the number of doses produced and potency assays (secretion of pro-inflammatory cytokine IFN after co-culture with tumor lines), transduction efficiency and phenotypic characterization were performed either in process or on the final product. The results suggest that the CAR-T cells produced have a high specific anti-tumor reactivity towards CD19 positive cells based on IFN-γ secretion.

These two examples highlight that in-house academic production is at least feasible for small cohorts of patients, with the achievement of clinical responses. Indeed, the study carried out by the Sheba Medical Centre showed that, 28 days after CAR-T cells infusion, 15 of the 20 patients who received CAR-T cells were minimal residual disease (MRD) negative, 3 were MRD positive and 2 patients had died from disease progression [23,24]. The study carried out at the Hospital Clínic de Barcelona, Spain showed a complete response rate of 71.1% at day +100 and an overall survival rate of 68.6% at 1 year [25].

There was almost no product out of specifications, which illustrates that these two processes are robust and reproducible. Moreover, in-house production allows (i) the infusion of fresh CAR-T cells, and (ii) a short delay between collection and infusion of CAR-T cells. In practice, these two examples illustrate the inherent complexity and diversity of ATMP manufacturing. As mentioned before, the scale out to automated devices is paving the way to a harmonization of production practices, which is necessary to analyze the results generated by academic centers. However, we must keep in mind that these devices are restricted to a single product for one patient in each production run, limiting the widespread application of CAR-T cell therapy. Nevertheless, as mentioned before, dedicated production for one patient is a well-known process in academic centers, which are implicated in HSCT. This local production is thus demonstrated in these two examples, which could also show a financial benefit. A decentralized local academic production would be much less expensive than that of the pharmaceutical companies. Indeed, the list price is between USD 370,000 to 475,000 for marketed specialties in the USA. The cost of the academic production of CAR-T cells has been estimated to be ten times less expensive than industrially-produced CAR-T cells [26].

## 4. Discussion

The overview of international trials up to 2022 showed that most of the trials are in recruitment or active status with a majority of phases I and II trials, even 20 years after the first patient infusion. As seen in the analysis, clinical trials for CAR-T cells are mainly carried out by USA and China, which remain the two leaders in terms of the number of trials, variety of conditions and targets studied. On the other hand, European trials on CAR-T cells are still very limited. We were able to see that Europe also has a large majority of trials in phase I and phase II. One advanced phase trial (Phase III) was led by Novartis Pharmaceutical (Basel, Switzerland) was suspended. Moreover, the European trials are sponsored by either an academic center or by a pharmaceutical company, and the collaboration between the two is still very weak (three European trials), whereas the latter represents most cases for international trials. The indications studied relate mainly to hematological malignancies, especially lymphoblastic leukemia and lymphoma, which were the first indications targeted by the treatment of CAR-T cells. For these hematological malignancies, CD19 remains the main target studied by European trials, although it has been widely studied for several decades. Moreover, even though there are struggles with solid tumor treatment in general, Europe gathers only 14 studies compared to 400 international studies, with just two innovative targets compared to international trials. From the data collected, we can see that Europe remains far behind the USA and China in this field in terms of the number of CAR-T cell trials. This observation could be explained at least partly by the difference in financial and regulatory policies dedicated to the development of these ATMPs. Indeed, the USA and China implement programs that facilitate the funding of research for innovative drugs.

### 4.1. Regulatory Differences

The American, European and Chinese regulatory agencies have their own specialized committees to evaluate advanced therapies [27]. With the development of ATMPs over the last 20 years, regulatory authorities have had to adapt to allow their marketing. collaboration between the FDA and the European Medicines Agency (EMA) has improved this regulation, despite differences still existing. These different regulatory authorities have different classifications for the qualification of ATMPs. In terms of marketing approval, there is specific legislation depending on the legal categorization of the product that can explain why some ATMPs are marketed in some regions but are not authorized in others, especially in Europe. China has also launched a policy of developing ATMPs with clear guidelines for rapid evolution [28]. The National Medical Products Administration (NMPA) of China wants to align with the ICH (International Council for Harmonisation of Technical Requirements for Pharmaceuticals for Human Use) guidelines. For clinical trials, a dual track regulation mode exists for the development of ATMPs. Clinical trials are typically initiated in investigator-initiated trial (IIT) mode in individual hospitals, before being transferred to the investigational new drugs (IND) file for marketing submission. Product registration from foreign data can be directly transposed in China for rare diseases and high-unmet medical needs, without running a trial. For high-unmet needs with a low prevalence in the population, a bridging study needs to be run with a limited patient number. The current regulatory requirement in China is less stringent than in the USA, requiring permission only from the internal hospital ethics committees to run a clinical trial on a cell and gene therapy product. This makes it easier to navigate the complex regulatory processes and get ATMPs into clinical trials, but it can also lead to the introduction and study of questionable products if there is not appropriate oversight and rigorous product validation criteria.

In recent years, regulatory agencies have launched various accelerated programs to reduce the processing time and enable products to quickly reach the market. ATMPs are generally eligible for these programs, which mainly concern products that fulfil an unmet medical need or have a potential major therapeutic benefit. The FDA has launched the “Fast Track Designation, Breakthrough therapy”, “Accelerated approval”, and “Priority review and Regenerative Medicine Advanced Therapy” programs. The NMPA encourage biotechnology innovation with reinforcement in the same vein, with “Breakthrough Therapy”, “Priority review”, and “Conditional approval” designations. For the EMA, we can mention the following programs: “Conditional Marketing Application”, “Authorisation under exceptional circumstances”, or “Accelerated assessment” [29]. In 2016, the Priority Medicines (PRIME) program allowed for the initiation of an early dialogue with the product developer in order to accelerate the process. The Breakthrough Therapy, Fast Track, and PRIME designation systems share the same objective of faster access to innovative medicines, but these programs have a different legal basis, making comparison and harmonization difficult [27]. In 2022, the EMA set up a pilot study to support non-profit academic organizations in the development of ATMPs. This pilot will provide enhanced regulatory support for up to five selected ATMPs that address unmet clinical needs and are only developed by academic and non-profit developers in Europe. These regulatory processes concern the best practice principles for manufacturing and clinical development planning that meets regulatory standards (regulatory flexibilities, development support measures, fee reductions). The first ATMP that will benefit from this pilot study is ARI-0001, developed by the Hospital Clínic de Barcelona [22], one of the most advanced centers for the academic production of CAR-T cells. This CAR-T cell product also previously received PRIME designation in 2021.

The question of intellectual property must also be clear for both parts, with the pharmaceutical company remaining the one to carry the patent until marketing authorization. In 2019, the Novartis group withdrew its patent on anti-CD19 CAR under pressure from the associations Médecins du Monde and Public Eye, thus allowing academic centers to study and manufacture anti-CD19 CAR-T cells. For several years, the Chinese government has been moving towards harmonization for the protection of intellectual property, which interests pharmaceutical companies looking to enter the Chinese ATMP market [30].

### 4.2. Financial Support

The three regions benefit from financial support to promote ATMP development, but not to the same extent. In the USA, the NIH, part of the U.S. Department of Health and Human Services government institutions, conducts medical and biomedical research. They generate numerous collaborations with academic and/or industrial centers for the development of CAR-T cells. Moreover, the U.S. Department of Health and Human Services is working with the Biomedical Advanced Research and Development Authority (BARDA), which benefits from a panel of industrial collaborators, allowing the support of ATMP development. This system was particularly highlighted during the COVID-19 pandemic, with a fast and efficient reaction in the production of vaccines, for example. In the same vein, China emphasizes the collaboration of multinational companies and the investment of capital in cell and gene therapies. This is evidenced by the number of clinical trials, although not all the information on these trials can be found on the clinical trials website. Collaborations between Chinese companies and international companies, such as Johnson & Johnson (New Brunswick, New Jersey), Roche (Basel, Switzerland), or Merck (Rahway, New Jersey), have allowed the manufacturing services in China to grow. The discussion of intellectual property is also progressing, making it possible to interest and promote companies seeking to engage in gene and cell therapy products. Within Europe, we have seen that collaborations with industrial groups are not yet on the agenda. The majority of trials are either carried out by an academic center or by a pharmaceutical company. The close collaboration of government organizations and industrial groups is poorly developed, which can notably explain this delay compared to the USA and China. The support of an industrial partner seems to be a necessity for academic centers, in order to pursue development towards advanced phases of clinical trials, as we have seen with the latest medicines approved on the market [31]. The implementation of policies supporting ATMPs is beginning to take shape in Europe. For example, in France, a bioproduction/biotherapy policy in the program France 2030 is making it possible to release investment funds in this area, encouraging collaborations between pharmaceutical companies and academic centers. Another example in Europe is the CARAMBA trial, which is led by a consortium of 11 partners from Germany, Italy, France and Spain, and pharmaceutical companies, notably DRK-Blutspendediens (Hessen, Germany), who possess 18 manufacturing sites in Germany. This project focuses on an anti-SLAMF7 CAR-T cell in multiple myeloma, which is manufactured virus-free [32]. This trial is part of the Horizon2020 research and innovation program on new therapies and rare diseases, and has been supported for over four years with funding of 6.1 million euros [33].

### 4.3. Academic Center for Autologous Production

As previously mentioned, the production of autologous CAR-T cells generates an intricate circuit, with several healthcare and industrial partners having to work together to ensure patient treatment and the management of side effects. This complexity leads to an increase in the raw cost of this therapy. Academic centers, which already manage both autologous and allogeneic HSCT for patients, could legitimately pretend to be CAR-T cell producers in an autologous setting. Indeed, through their proximity to hospitals, academic production centers allow translational research to be carried out, working directly on patient samples. It is possible to screen patients’ tumors to study potential new tumor targets, especially in solid tumors. The study and optimisation of several types of cell sources, different gene sequences and expansion conditions are generated primarily in these centers. The development of virus-free CAR transduction techniques (e.g., Sleeping beauty, CRISPR/Cas9) is also one of the current challenges in the manufacturing process [34,35]. These centers may also benefit from nearby GMP facilities approved by regulatory authorities, which allow the production of ATMPs for patients [36]. In France, the French-speaking society of bone marrow transplantation and cellular therapy (SFGM-TC) provides recommendations for the research and development of CAR-T cells by academic centers [37]. To date, clinical trials are mainly focused on an autologous cell source with a hospital and industrial circuit that is now well described, with preventive management of side effects [38]. The final product derived from a patient is dedicated to the same patient. Due to the complexity of CAR-T cell production, it would be easier to consider this type of production with the use of automated systems. These systems have been on the market for several years, the best known being the Miltenyi Prodigy^®^ (Bergisch Gladbach, Germany, and more recently the ADVA X3 from AdvaBio (Haïfa, Israel) [39]. In addition to CAR-T cell production, this last platform provides interesting flexibility in the choice of buffers and on-line monitoring throughout the process. It includes software that enables the programming and design of any predefined protocols, so it can provide a customized product for each individual patient. These new features, which were not available on previous devices, could facilitate point-of-care manufacturing for several patients. This device will be tested in a clinical trial for the first time in 2023. The development of fully automated production platforms with in line controls and advanced data analysis are currently being implemented to overcome these challenges [17,21].

### 4.4. Allogeneic CAR-T Cells

In the near future, the development of allogeneic CAR-T cells seems the best strategy to allow wide use of this therapy, provided immune tolerance is achieved. Nevertheless, studies are still ongoing to generate a product from a controlled cell source, off-the-shelf, reducing infusion delay and cost. This configuration seems to match better with pharmaceutical companies, with standardized procedures for the production and qualification of a batch. This would require scale-up production systems, with the possibility of on-line product controls and real-time feedback. It could allow production to be adapted to each raw material from different cell types, and quality control throughout the process. These controls remain heterogeneous tests, with different parameters followed due to the large number of clinical trials on CAR-T cells, and there is an absence of guidelines for the implementation of these controls [40]. Harmonization of these phenotype, expansion, functional, and potency tests would allow CAR-T cell controls to be aligned on the same basis, improving the final comparability of CAR-T cell characteristics before release. It would also provide more important manufacturing, storage and logistic capacities than can be achieved by an academic center.

The scale-up of allogeneic CAR-T cell production in bioreactors will allow the generation of batches containing a high number of specific cells [41,42]. The product could be followed throughout the process with automated and on-line monitoring systems, with the possibility of feedback, while remaining in a closed system. Indeed, CAR-T cells remain a living cellular product with inter-individual variability. This has an influence on the quality of the final product; being able to adapt the state of the intermediate product and modify the parameters would be ideal to improve the final yield, i.e., the number of cells and their viability. The development of these programs is in the near future, with the advent of “machine learning”. Moreover, manufacturers would benefit from a digital twin in silico, which would also allow them to perform upstream simulations on the variation of a parameter during production. This in-line control and feedback would save time and money, with a better knowledge of intermediate and final products. The implementation of this new kind of software, based on artificial intelligence, would allow the production of ATMPs to evolve to a new stage [21]. This therapy is undergoing changes in its conception and in the management of its academic and industrial production. Regulatory evolution concerning ATMPs and financial support from each country would also allow solutions to be found to accelerate their development and market authorization, especially in Europe.

## 5. Conclusions

The development of the first CAR-T cells started in an academic center, before the establishment of industrial collaborations, leading the product towards marketing authorization. Four new CAR-T cell ATMPs have recently been authorized, all the result of collaborations, showing that this alliance is still necessary to bring this therapy to this crucial stage. Europe, which lags behind the USA and China, has not put forward this kind of collaboration.

However, industrial production implies a high cost, with restrictive logistics, especially for the patient and the agents involved in the hospital circuit. It is conceivable that academic centers could take over the production of autologous CAR-T cells. Indeed, academic centers remain the primary players in the development of this therapy. Efforts by regulatory authorities are beginning to emerge to fill the unmet gaps that remain in anti-cancer therapy, notably in Europe. These efforts should support academic centers efforts to bring their ATMPs to the market, with regulatory requirements that match the capacity of an academic center. We have seen that Europe is lagging behind, probably due to the low level of academic-industry collaboration, which has not been encouraged by financial support and is hindered by unfavorable regulations. The development of CAR-T cells should certainly evolve towards collaborations based on the USA model, while providing support to academic centers who could use their experience to participate in the production of autologous CAR-T cells. In France, considering public institutions are not legally authorized to produce and sell pharmaceutical products, the participation of academic centers is still limited to clinical trials. This does not apply to other European countries, who could rely on their academic centers with an organization model that could reduce production costs and thus the cost of a CAR-T cell product.

The delegation of production to academic centers, under the cover of an industrial partnership or the creation of a spin-off from the academic center, would make it possible to reduce this cost, while industry could focus on the production of allogeneic CAR-T cells, known as off-the-shelf.

## Figures and Tables

**Figure 1 cancers-15-01003-f001:**
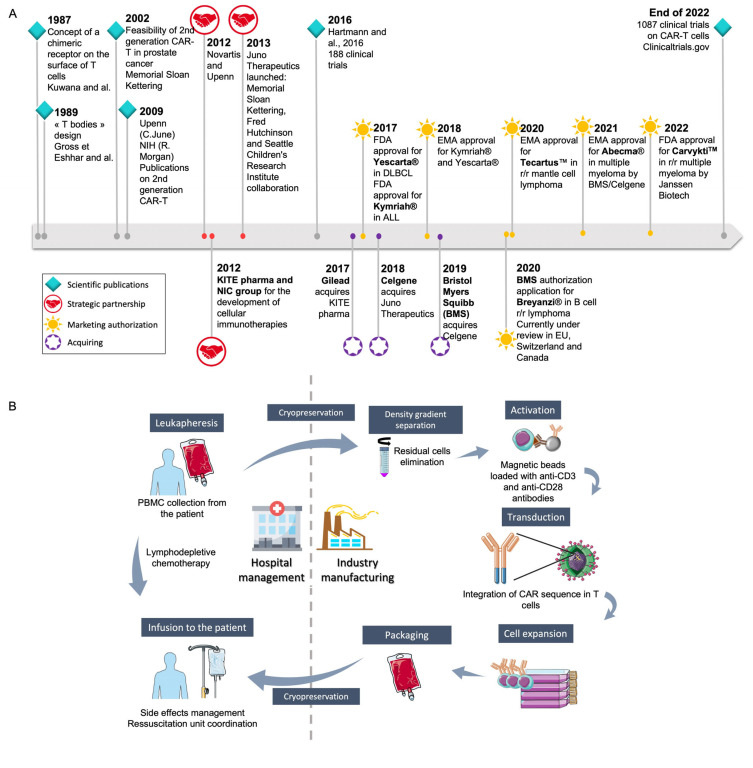
Generalities on CAR-T cell development and their industrial and hospital management. (**A**) The way to marketing authorization of the different available CAR-T cells: academic discoveries led either (i) to start-up creation (Juno Therapeutics, Celgene, Kite pharma) to allow the development of clinical trials, before getting acquired by pharmaceutical companies or (ii) to the licensing to a pharmaceutical company. CAR: Chimeric Antigen Receptor; DLBCL: Diffuse Large B-Cell Lymphoma; EMA: European Medicines agency; FDA: Food and Drug Administration; NIC: National Cancer Institute; NIH: National Institutes of Health; r/r: relapsed refractory [2,10,11] (**B**) The logistic to allow the production of autologous CAR-T cells by pharmaceutical companies. Unlike other medicinal products, the raw material is specific to each CAR-T cell production. This required the participation of academic centers on crucial step such as collection of the leukapheresis or preservation of the ATMP and thawing before infusion into the patient. Different steps are not under the pharmaceutical company control, unless an audit validates the compliance with the company procedures.

**Figure 2 cancers-15-01003-f002:**
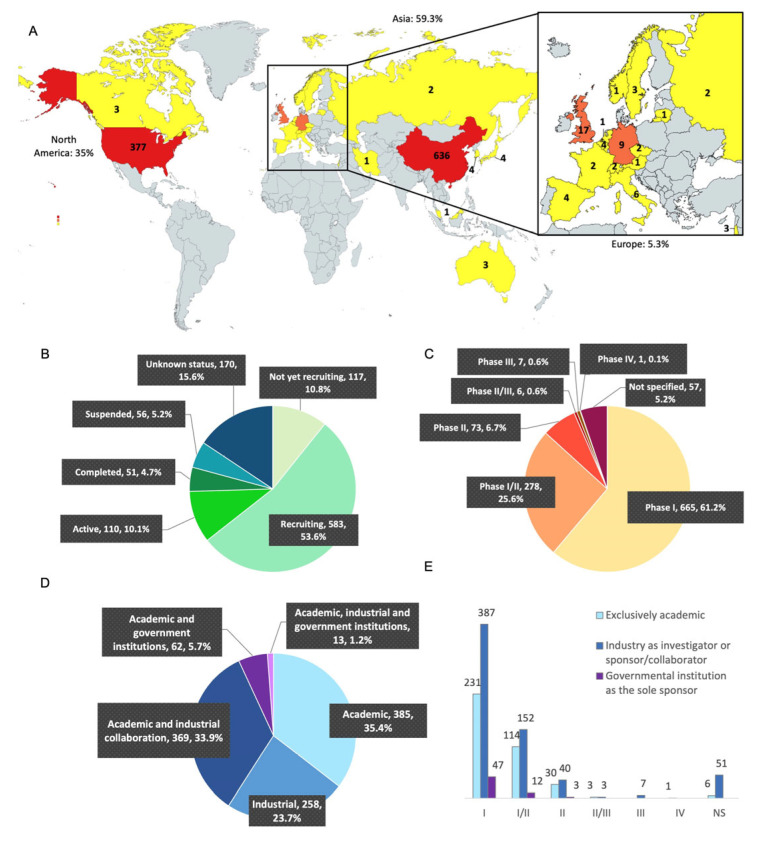
Distribution of CAR-T cell clinical trials in the world in 2022. (**A**) Geographical distribution: The United States and China dominate in number of studies, with 377 (34.7%) and 636 (58.5%) studies, respectively. North America is also represented by Canada (3 studies). Apart from China, the Asian countries have 9 studies (0.8%) with Japan (4), South Korea (4), Malaysia (1) and Iran (1). Europe has 58 studies (5.3%): UK (17), Germany (9), Italy (6), Belgium (4), Spain (4), Sweden (3), Israel (3), Switzerland (2), Czech Republic (2), Russia (2), France (2), Lithuania (1), Netherlands (1), Austria (1), and Finland (1). Mapchart.net. (**B**) Status distribution: 117 studies have not started the recruitment (10.8%), 583 are currently recruiting patients (53.6%), 110 are active (10.1%), 51 have been completed (4.7%), 56 have been discontinued (5.2%), and 170 have unknown status (15.6%). (**C**) Study phases: 665 studies are in phase I (61.2%), 278 in phase I/II (25.6%), 73 in phase II (6.7%), 6 in phase II/III (0.6%), and 7 in phase III (0.6%). One study is in phase IV (0.1%) and 57 studies did not mention the information (5.2%). NS = not specified. (**D**) Distribution of clinical trials according to investigators and collaborators: out of 1087 clinical trials, 385 are mentioned only with an academic institution (35.4%), 258 are carried out only by a pharmaceutical industry (23.7%), 369 are carried out by an academic center in collaboration with an industrial company (33.9%), 62 are carried out by an academic center in collaboration with a governmental institution such as the NIH (5.7%), and finally 13 of the trials are carried out by a collaboration between these three actors (1.2%). (**E**) Distribution of clinical trials according to investigators and collaborators: number of academic clinical trials or in collaboration with an industry or in collaboration with a governmental institution according to the phases. The trials led by the three actors at the same time were added to the industrial collaboration group.

**Figure 3 cancers-15-01003-f003:**
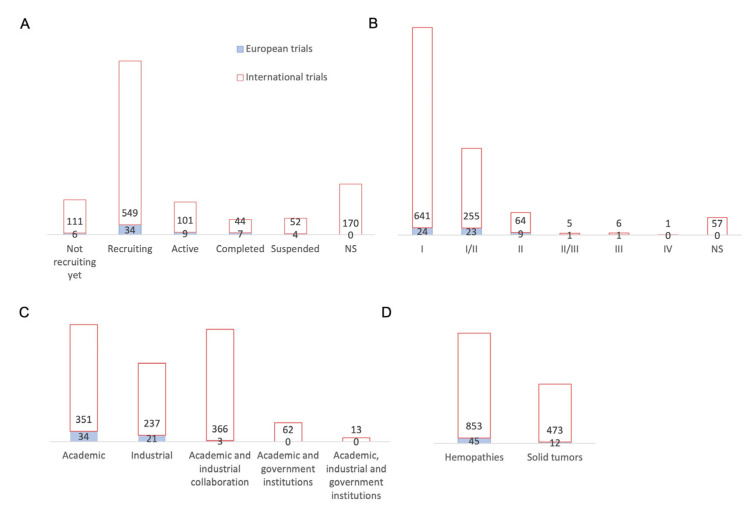
Comparison of European CAR-T cell clinical trials compared with international trials. (**A**) Status distribution: 6 studies have not started the recruitment, 34 are currently recruiting patients, 9 are active, 7 have been completed, and 4 have been discontinued. (**B**) Study phases: 24 studies are in phase I, 23 in phase I/II, 9 in phase II, 1 in phase II/III, and 1 study in phase III. (**C**) Distribution of clinical trials according to investigators and collaborators: out of 58 clinical trials: 34 are carried out by an academic center, 21 are carried out by a pharmaceutical industry, and 3 are carried out by an academic center in collaboration with an pharmaceutical company. (**D**) Distribution of conditions: hematological malignancies were studied in 45 European studies and solid tumors in 12 European studies. If we compare with international trials outside Europe, we can see that hematological malignancies were studied in 853 studies and solid tumors in 473 studies. In this graph, studies can target multiple conditions, which explains the greater number of conditions investigated compared to the number of studies. NS = not specified.

**Figure 4 cancers-15-01003-f004:**
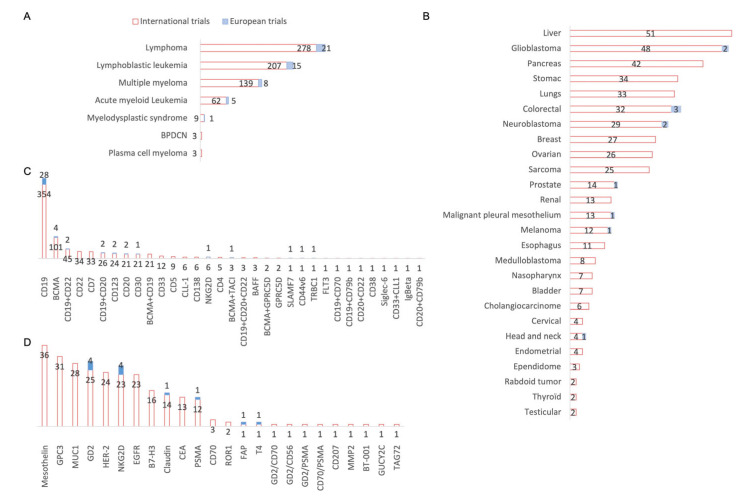
Conditions and targets studied in international and European CAR-T cells clinical trials. (**A**) Focus on hematologic malignancies: 21 European studies focus on lymphomas all combined, 15 studies on lymphoblastic leukemia, 8 studies on multiple myeloma and 5 studies on acute myeloid. BPDCN: Blastic plasmacytoid dendritic cell neoplasm. (**B**) Focus on solid tumors: 3 European studies focus on colorectal cancer, 2 studies on each glioblastoma and neuroblastoma and one study focuses on prostate cancer, malignant pleural mesothelium, melanoma and head and neck cancers. (**C**) Focus on targets investigated in hematologic malignancies: 28 European studies focus on CD19 targeting, 4 studies on BCMA and the following are investigated in 2 studies: CD123, CD20, CD30, dual targeting of CD19 combined with CD22 and CD19 combined with CD20. Finally, one study is investigating NKG2D, SLAMF7, TRBC1 and CD44V6 alone and the dual targeting of BCMA and TACI. These targets were studied in 45 European studies of 1087 international studies of hematological malignancies. BAFF: B-cell activating factor; BCMA: B-cell maturation antigen; CD: Cluster of differenciation; CLL-1: C-type lectin-like molecule-1; FLT3: FMS-like tyrosine kinase 3; GPRC5D: G protein–coupled receptor, class C, group 5, member D; Ig: Immunoglobulin; NKG2D: Natural Killer Group 2 membrane D; SLAMF7: Signaling Lymphocytic Activation Molecule Family Member 7; TACI: Transmembrane activator and Calcium modulating ligand interactor; TRBC1: T-cell Receptor Constant β Chain-1. (**D**) Focus on targets investigated in solid tumors: 4 European studies are investigating NKG2D and GD2, and the following targets are investigated in 1 study: CLDN6/Claudin18.2, PSMA, FAP and T4. These targets were studied in 12 European studies of 221 international solid tumor studies. B7-H3: B7 homolog 3 protein; BT-001: not specified; CD: Cluster of Differenciation; CEA: Carcinoembryonic Antigen; EGFR: Epidermal Growth Factor Receptor; FAP: Fibroblast Activation Protein Alpha; GD2: Disialoganglioside; GPC3: Glypican-3; GUCY2C: Guanylate Cyclase 2C; HER-2: Human Epidermal Growth Factor Receptor-2; MUC-1: Mucin 1; MMP-2: Matrix Metalloproteinase-2; NKG2D: Natural Killer Group 2 membrane D; PSMA: Prostate-Specific Membrane Antigen; ROR-1: Receptor Tyrosine Kinase Like Orphan Receptor 1; T4: Thyroxin 4; TAG72: Tumor-associated glycoprotein 72.

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
