# Peer review of "Systematic Review on CAR-T Cell Clinical Trials Up to 2022: Academic Center Input"

_cancers, 2023, doi:10.3390/cancers15041003_

Round 1

Reviewer 1 Report

The article entitled "Retrospective analysis of CAR-T cell clinical trials up to 2021: the role of academic centers" by Wang V et al is a really interesting review about the level of development of CAR-T around the world and the role of academic centers and pharma companies in the clinical trials and the final market authorization. Authors develop a very important analysis for understanding how the future of this type of treatment in the world and specially in Europe (where the cost and the final accessibility to patients is not clearly guaranteed) will be. But for an article that should be published in 2023 it does not seem acceptable that the data were collected in 2021. This referee understands that it is very difficult now to re-collect and reanalyze 2022 data again, but at least re-collect some of them in 2022 should be "a must", ... otherwise it does not make a complete sense to publish this article in 2023 (and it merits to be published!).). Maybe, while not all analyses should be redone by authors, at least authors could incorporate some new data from 2022 into the discussion, to demonstrate that clinical trials in 2022 do not substantially change the conclusions of the paper. This is obviously the major concern for this reviewer.

Other concerns:

- Self-citations in reference 14 (Qian C et al. "Curative or pre-emptive adenovirus-specific T cell transfer .... J Hematol Oncol. 2017, 10(1), 102) is a self-citation that it is unnecessary, or at least forced, ...  if not inappropriate.

- The use of "(n)" for specify a number of cases for a characteristic is misleading in the article, because it is the same way to indicate references. Authors maybe can use "[n]".

- Before (or after) the indication that in 2022, the EMA has set up a pilot study to support non-profit academic organizations developing ATMPs, maybe authors could indicate that "PRIME designation" was granted in december of 2021 to the first academic drug (ARI-0001), being this first designation, granted to this CAR-T product (https://www.ema.europa.eu/en/human-regulatory/research-development/prime-priority-medicines#list-of-products-granted-eligibility-section).

Author Response

Thank you very much for your kind words about this review. Here are the responses to your comments. I have a question for the Response 3.

Point 1: The article entitled "Retrospective analysis of CAR-T cell clinical trials up to 2021: the role of academic centers" by Wang V et al is a really interesting review about the level of development of CAR-T around the world and the role of academic centers and pharma companies in the clinical trials and the final market authorization. Authors develop a very important analysis for understanding how the future of this type of treatment in the world and specially in Europe (where the cost and the final accessibility to patients is not clearly guaranteed) will be. But for an article that should be published in 2023 it does not seem acceptable that the data were collected in 2021. This referee understands that it is very difficult now to re-collect and reanalyze 2022 data again, but at least re-collect some of them in 2022 should be "a must", ... otherwise it does not make a complete sense to publish this article in 2023 (and it merits to be published!).). Maybe, while not all analyses should be redone by authors, at least authors could incorporate some new data from 2022 into the discussion, to demonstrate that clinical trials in 2022 do not substantially change the conclusions of the paper. This is obviously the major concern for this reviewer.

Response 1: Sorting and analysis of the 2022 data is currently underway and will be implemented upon correction of the review.

Other concerns:

- Self-citations in reference 14 (Qian C et al. "Curative or pre-emptive adenovirus-specific T cell transfer .... J Hematol Oncol. 2017, 10(1), 102) is a self-citation that it is unnecessary, or at least forced, ...  if not inappropriate.

Response 2:  This citation and the sentence about the work of our team will be deleted. 

- The use of "(n)" for specify a number of cases for a characteristic is misleading in the article, because it is the same way to indicate references. Authors maybe can use "[n]".

Response 3: References are in square brackets, and numbers of clinical trials are in parentheses. If everything is not in order, can you tell me what needs to be corrected? Or do you think I should add "n=" in each parenthesis to list the trials?  

- Before (or after) the indication that in 2022, the EMA has set up a pilot study to support non-profit academic organizations developing ATMPs, maybe authors could indicate that "PRIME designation" was granted in december of 2021 to the first academic drug (ARI-0001), being this first designation, granted to this CAR-T product (https://www.ema.europa.eu/en/human-regulatory/research-development/prime-priority-medicines#list-of-products-granted-eligibility-section).

Response 4: The following sentence will be added just after the indication: "This CAR-T also previously received PRIME designation in 2021."

Reviewer 2 Report

1. Improve the resolution of all figures 

2. Authors should follow the English style of writing and replace commas with periods.

3. Bold the figure numbers inside the caption for better readability. There may be no need of putting 4A, 4B, 4C...inside the caption, just A, B, C with bold style will be fine. 

4. The sentences should be short and complete for better readability. 

Author Response

Thank you for your comments. Here are the responses to your comments. I have a question for the response 1.

1. Improve the resolution of all figures 

Response 1: The resolution of all the figures reach 330 dpi (>300dpi), 2135pixels width/height at least (>1000), RGB at 8 according to the guidelines. I inserted the figures in the text on a word document by inserting an image from a file. This method may have decreased the resolution of the figures. Can you tell me how to proceed to maintain the resolution of the figures? 

2. Authors should follow the English style of writing and replace commas with periods.

Response 2: The corrected version of the review will be edited by a translator certified for the publication of scientific articles. It is ongoing. I will replace commas with periods.

3. Bold the figure numbers inside the caption for better readability. There may be no need of putting 4A, 4B, 4C...inside the caption, just A, B, C with bold style will be fine. 

Response 3: I will put the figure numbers inside the caption in bold and I will only leave the numbers without letters.

4. The sentences should be short and complete for better readability. 

Response 4: The generation of short and complete sentences was requested from the translator.

Round 2

Reviewer 1 Report

The new version give responses to all my concerns.